# Impact of Transportation Costs on the Establishment of an Industrial Symbiosis Network

**Mohamed Amine Anane, Faezeh Bagheri, Elvezia Maria Cepolina *** and **Flavio Tonelli**

Department of Mechanical, Energy, Management and Transport Engineering (DIME), University of Genoa, 16145 Genoa, Italy; amineanane22@gmail.com (M.A.A.); faezeh.baagheri@gmail.com (F.B.); flavio.tonelli@unige.it (F.T.)
* Correspondence: elvezia.maria.cepolina@unige.it

**Abstract:** The challenges related to natural resource depletion and environmental issues stimulate businesses to look for solutions to overcome them. One of the leading strategies that have emerged from the practical implementation of the circular economy concept is industrial symbiosis, which aims to reduce material extraction and consumption by using the waste (co-product) of one company as input for production processes of another company. This study aims to provide a more profound insight into industrial symbiosis (IS) modeling by considering the transport system impact. To this end, a hybrid approach based on agent-based modeling and system dynamics is presented to comprehensively capture the complexity of interactions between companies and their related impacts on transportation. A case study and numerical example are discussed to validate the proposed approach and related model. The results demonstrate that the development of IS, as expected, is significantly influenced by the transport system.

**Keywords:** sustainability; industrial symbiosis; transportation; system dynamics; agent-based modeling

## 1. Introduction

Industrial symbiosis (IS) is a reconfiguration of the production network from the heart of the circular economy. The underlying strategy of this concept pushes companies to collaborate with each other and take advantage of resources, from physical resources such as raw materials, energy, waste, etc., to non-substantial resources including knowledge, skills, etc. [1]. The proposed research focuses on the exchange of physical resources [2,3]; the waste or by-products of one industry or industrial process could be considered as raw material for another company. Waste is considered a non-value-added material, and factory and company owners are being pushed to invest resources to get rid of it. This means that waste represents an additional financial burden for companies and reduces profits. In this context, IS could be a production strategy that can handle this challenge more efficiently.

Although operating an IS allows for a reduction in manufacturing costs [4,5], according to Yazan et al. [6], it also imposes three additional charges on enterprises: (a) costs associated with waste transportation, which involves moving waste from the producer to the user; (b) costs associated with waste treatment, which involves preparing wastes for use as inputs; and (c) costs associated with transaction costs, which are associated with managing and coordinating the IS business [7].

So far, there has been scant attention in the literature to address the role of transportation costs in an industrial symbiosis business model. To the best of our knowledge, only one article [6] attempted to perform it.

There are no studies that investigate the impact of different transport modes on costs associated with waste transportation, which in turn affects IS feasibility. In fact, waste can be transported using different modes of transportation, and each mode has its own cost and to its own impact on the environment.

This paper is aimed at filling this gap by developing a new hybrid model, that includes agent-based and system dynamics approaches, to test the feasibility of the IS business model through several scenarios based on different transportation systems. The choice of the hybrid model for modeling the system under study is novel and justified by the findings of Demartini et al. [8]. Indeed, the hybrid approach can capture both the detailed and dynamic complexity of IS: it involves multiple domains and agents and displays non-linear and non-rational interacting behaviors [9–11]. Specifically, the paper proposes a hybrid model that makes it possible to model production processes synthetically, via an SD approach, according to Wang et al. [12] and Norbert et al. [13], and industrial plants via an AB approach, according to Cui et al. [14–18]. The AB approach makes it possible to model the following three main aspects of the system under study:

1.  Agent-plants have a behavior in establishing IS relationships that is determined by shared collective values and rules. The available transport systems affect them.
2.  Shared collective values and rules evolve because of the interaction between agent-plants.
3.  Agent-plants cannot be considered equal or similar, but each has its own status which is determined by the functioning of internal production processes and modeled with SD logic, and plays a specific role.

The proposed model was developed to analyze the specific case of possible symbiosis between steel plants (SPs) and cement plants (CPs). A review of successful industrial symbiosis case studies in the cement industry was performed by Krese et al. [19]. Regarding industrial symbiosis in the steel sector, interesting information is provided by Branca et al. [20]. Their analysis shows that the use of steel and blast furnace slag in the production of cement is the most common synergistic exchange [19].

The aim of the proposed research is to test, using a hybrid approach, in a network of SPs and CPs, how IS relationships evolve over the long term, considering different transport systems.

The rest of the paper is organized as follows. Section 2 presents the materials and methods. Section 3 introduces the hybrid model, the key performance parameters, and the simulated scenarios. Section 4 provides the computational results of the hybrid model, and proposes a sensitivity analysis of the system dynamic component. Finally, Section 5 concludes the paper.

## 2. Materials and Methods

The waste from an SP plant could be used by CP. In this case, let us assume that two types of raw materials, including ferrous scrap and carbon coke, are required for producing steel. The raw materials required for the CP plant are clinker and natural inert material. In this context, the waste produced by SP (steel and blast furnace slag) could be used by CP as an artificial inert material instead of a natural inert material (Figure 1).

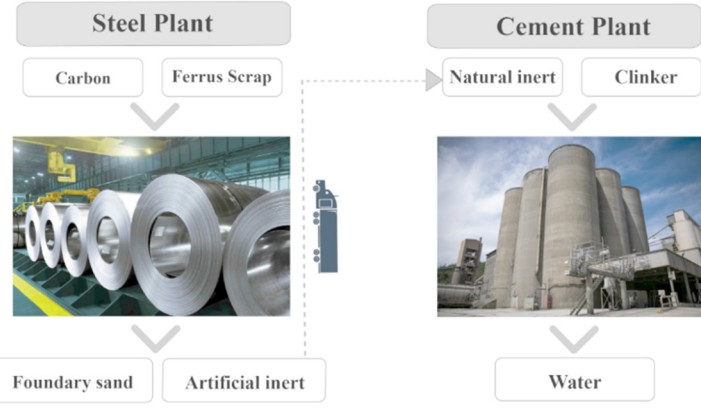

**Figure 1.** The link between SP and CP.

Reusing slag from the manufacturing process of steel not only enhances the properties of cement, such as durability due to a longer setting time, but it also has significant positive effects on the economy and the environment because it uses up to 80% less energy than it would to produce regular Portland cement [19]. Eco-indicator 99 for natural and artificial inert materials [21] showed that the production of natural inert materials has negative consequences on human health (respiratory problems, radiations, carcinogens) and on ecosystem quality (eco-toxicity, acidification). It simultaneously leads to the massive exploitation of mineral and fossil resources. However, many studies [22,23] have revealed that artificial inert materials generate a consistent impact on ionizing radiation and land use. Therefore, there is a need to consider that materials should be stored before being recycled. Re-processing inert materials from waste allows other factories to obtain the same product as natural inert materials, but with a lower environmental footprint and the negative effects on human health, despite the need for land to store the waste before being processed. Nevertheless, the extraction of material from land generates much higher impacts, so the most sustainable choice (whenever possible) is the symbiosis between the two industries.

The framework of the proposed IS model is defined through a series of general assumptions presented below:

- A1: The network includes SPs and CPs that could share mutually beneficial transactions.
- A2: The mentioned factories are in the same country and on the same land to avoid problems related to sea transportation and changes in juridical requirements.
- A3: Waste of one factory (SP) is considered an input raw material of another factory (CP).
- A4: Transport between the two plants can be performed by a single-modal transport system (road transport by truck) or with a bi-modal transport system where two means of transport (roadway and railway) are used by the same operator on different routes and at different times of the day (the vehicles involved are trucks and trains).
- A5: The factory, which is the source of waste (SP), financially supports another factory (CP) to take and utilize its waste.
- A6: The transformation costs are in charge of the CP that uses the waste as a raw material.

## 3. Hybrid Models

The proposed modeling adopts AB to model the learning and complex behaviors of CPs and SPs within the IS network and SD to model supply chains and the production flows of each plant. The combination of SD and AB, which captures heterogeneities, enables us to understand the complex dynamics of IS.

SPs and CPs are nodes of a network and are agents: SP nodes are waste origin nodes, whereas CP nodes are potential waste destination nodes in the IS network. Links connect SP nodes to CP nodes. A link represents an IS relationship between its origin node and its destination node. We assume that an SP node can be the origin of several links while a CP node is the destination of only one link, meaning that an SP can send its waste to several CPs while a CP can use waste from only one SP. This IS relationship is feasible under certain conditions in each time window. The IS relationship will be feasible if there is convenience for both the CP origin node/agent and the SP destination node/agent under specific conditions. Each IS relationship with feasible results under specific conditions in the given time window corresponds to an amount of waste material that, in the given time window, is produced by the source node SP and that the CP node decides to use as a raw material and to the related values for key performance indicators (KPIs). The amount of waste material for each SP-CP link depends only on the production rate of the SP source node and the raw material demand of the CP destination node.

Tables 1 and 2 show the model parameters. Table 1 is related to SP, which is interested in removing its waste, while Table 2 is related to CP, which could use the waste as the raw

material (amount of symbiosis). Table 3 refers to the transport system used to move the waste material from the SP to the CP. A transport system may involve a single means of transportation (truck) or two means of transportation (truck and train). The values of the parameters for SP and CP are reported in Tables A1 and A2 in Appendix A.

**Table 1.** Model parameter for the SP.

| Parameter Description | Parameter Name | Type of Unit |
|---|---|---|
| Weekly demand | $WK\_demand\_SP$ | Mass |
| The amount of raw material required to produce one unit of the final product | $RM\_amount\_SP$ | Coefficient |
| Production capacity | $PC\_SP$ | Mass/Time |
| Price of raw material | $RM\_price\_SP$ | Currency/Mass |
| Amount of generated waste per unit of the final product | $FG\_waste\_SP$ | Coefficient |
| Sales price of the final product | $FG\_price\_SP$ | Currency/Mass |
| Landfill tax per unit of waste mass | $Unit\_Landfill\_Tax$ | Currency/Mass |
| Landfill tax paid in one week | $Landfill\_Tax$ | Currency |
| Economic contribution for the mass unit that SP pays to CP | $Unit\_Money\_From\_SP\_To\_CP$ | Currency/Mass |
| Economic contribution that the SP weekly pays to the CP | $Money\_From\_SP\_To\_CP$ | Currency |
| Other weekly costs (Labor, energy, maintenance, storage, etc.) | $WK\_othercosts\_SP$ | Currency |
| Weekly storage cost per unit of mass stored | $Unit\_Storage\_Cost$ | Currency/Mass |
| Waste quantity weekly produced by SP | $Waste\_Inventory\_Stock\_SP$ | Mass |
| Amount of profit that SP weekly obtains through IS | $Profit\_From\_Symbiosis\_SP$ | Currency |

**Table 2.** Model parameter for the CP.

| Parameter Description | Parameter Name | Type of Unit |
|---|---|---|
| Weekly demand | $WK\_Demand\_CP$ | Mass |
| The amount of raw material required to produce one unit of the final product | $RM\_amount\_SP$ | Coefficient |
| Production capacity | $PC\_CP$ | Mass/Time |
| Raw material price per unit of mass | $Unit\_Natural\_Inert\_Cost$ | Currency/Mass |
| Raw material price paid in one week | $Natural\_Inert\_Cost$ | Currency |
| Amount of generated waste per unit of the final product | $FG\_waste\_CP$ | Coefficient |
| Sales price of the final product | $FG\_price\_CP$ | Currency/Mass |
| Preprocessing cost per unit of mass | $Unit\_Preprocessing\_Cost$ | Currency/Mass |
| Preprocessing cost paid in one week | $Preprocessing\_Cost$ | Currency |
| The amount of inert material required for weekly production | $Natural\_Inert\_Order$ | Mass |
| Amount of inert material that CP must buy weekly from the supplier | $Natural\_Inert\_Supplier$ | Mass |
| Other weekly costs (Labor, energy, maintenance, storage, etc.) | $WK\_othercosts\_CP$ | Currency |
| Amount of profit that CP weekly obtains through IS | $Profit\_From\_Symbiosis\_CP$ | Currency |

**Table 3.** Model parameter for the m-Transport System and the v-vehicle.

| Parameter Description | Parameter Name | Type of Unit |
|---|---|---|
| Weekly amount of waste material produced by SP and that CP decides to use as raw material. Waste that is transported weekly from SP to CP by the m-transport system | $Symbiosis\_Amount$ | Mass |
| Transport cost related to 1 km travelled per load unit by the v-vehicle | $Unit\_Transport\_Cost_v$ | Currency/(Distance × Mass) |
| Cost of a trip by an unloaded v-vehicle | $Empty\_Trip\_Cost_v$ | Currency/Trip |
| Loading capacity of the v-vehicle | $Capacity_v$ | Mass |
| Length of the one-way route taken by v-vehicle between SP and CP | $Trip\_Lenght\_Km_v$ | km/Trip |
| kms weekly travelled by the v-vehicle | $Travelled\_Km_v$ | km |
| Tonnes of material weekly transported by v-vehicle | $Travelled\_Tons_v$ | Mass |
| Grams of $CO_2$ produced by the v-vehicle per unit of mass transported per 1 km | $\left(\frac{g\,CO_2}{ton \times Km}\right)_v$ | Mass/(Mass × Distance) |
| Weekly grammes of $CO_2$ produced by the m-transport system | $CO_2\_Emissions_m$ | Mass |
| Weekly one-way transport cost to bring material from the SP to the CP nodes using the m-transport system | $Transport\_Cost\_Week\_Load\_Trips_m$ | Currency |
| Weekly transport costs for empty return trips from CP nodes to SPs by the m-transport system | $Transport\_Cost\_Week\_Return\_Trips_m$ | Currency |
| Total weekly transport cost to move waste from SP to CP nodes, including return trips by the m-transport system | $Transport\_Cost_m$ | Currency |

In the following Section, the time window is considered equal to one week.

The conditions under which the IS relationship may be feasible concern the following: the economic contribution for each unit of mass that the SP node agrees to make to the CP node and the transport cost that depends on the transport system used to move the material between the origin node and the destination node. These conditions are iteratively modified, as described in Section 3.3, to find the optimal conditions for which the IS relationship is ultimately feasible.

Every time period, there are two alternatives for SP to get rid of its waste:

a.　SP could pay the landfill tax($-Landfill\_Tax$) and conduct landfilling or

b.　SP could give money to $CP(-Money\_From\_SP\_To\_CP)$ to take its waste and replace the natural inert materials with it. This amount of money is assessed according to (1)

$$Money\_From\_SP\_To\_CP = Unit\_Money\_From\_SP\_To\_CP \times Symbiosis\_Amount \tag{1}$$

This amount of money should cover the expenses (*Transformation_Costs*) supported by CP to use, as raw material, the waste from SP during the time period. *Transformation_Costs* include (2), (3), and (4):

$$Preprocessing\_Cost = Unit\_preprocessing\_Cost \times Symbiosis\_Amount \tag{2}$$

$$Transport\_Cost_m = Transport\_Cost\_Week\_Load\_Trips_m + Transport\_Cost\_Week\_Return\_Trips_m \tag{3}$$

where the two addends are described in Equations (3a) and (3b)

$$Transport\_Cost\_Week\_Load\_Trips_m = \sum_{\forall\, v \in m} Unit\_Transport\_Cost_v \times Travelled_{Kmv} \times Symbiosis\_Amount \tag{3a}$$

where the summation is extended to all vehicles $v$ in the transport system $m$

$$Transport\_Cost\_Week\_Return\_Trips_m = \sum_{\forall\, v \in m} \left(\frac{Symbiosis\_Amount}{Capacity_v}\right) \times Empty\_Trip\_Cost_v) \tag{3b}$$

where the summation is extended to all vehicles $v$ in the transport system $m$

$$Storage\ cost = Unit\_Storage\_Cost \times Symbiosis\_Amount \tag{4}$$

SP selects the less costly alternative.

Every time period, there are two alternatives for CP as well:

- place its order to purchase the natural inert materials from the natural inert materials supplier, according to Equation (5)

$$Natural\_Inert\_Supplier = Natural\_Inert\_Order \tag{5}$$

- or accept the SP offer ($+Money\_From\_SP\_To\_CP$), pay for the expenses ($Transformation\_Costs$), and purchase only the residual amount of inert materials, if any, from the natural inert materials supplier, according to Equation (6)

$$Natural\_Inert\_Supplier = Natural\_Inert\_Order - Symbiosis\_Amount \tag{6}$$

CP selects the alternative b if:

$$Transformation\_Costs - Money\_From\_SP\_To\_CP < Natural\_Inert\_Cost \tag{7}$$

where

$$Natural\_Inert\_Cost = Unit\_Natural\_Inert\_Cost \times Symbiosis\_Amount$$

The IS relationship will be feasible, under specific conditions, if there is convenience for both the origin CP node/agent and the SP destination node/agent. This occurs when both conditions (8) (SP's point of view) and (9) (CP's point of view) are met:

$$Money\_From\_SP\_To\_CP < Landfill\_Tax = Unit\_Landfill\_Tax \times Symbiosis\_Amount \tag{8}$$

$$Money\_From\_SP\_To\_CP > Transformation\_Costs - Natural\_Inert\_Cost \tag{9}$$

The feasibility check of the IS relationship is classified as an event. It is cyclic and repeated every time period.

If the IS relationship is feasible, Equations (10) and (11) will be established; otherwise, Equation (12) takes place:

$$Symbiosis\_Amount = \mathrm{MIN}(Waste\_Inventory\_Stock\_SP, Natural\_Inert\_Order) \tag{10}$$

$$Natural\_Inert\_Supplier = Natural\_Inert\_Order - Symbiosis\_Amount \tag{11}$$

$$Natural\_Inert\_Supplier = Natural\_Inert\_Order \tag{12}$$

The feasibility check of the IS relationships, which has been described in detail above, is part of a more complex model to simulate the complete supply chains of CPs and SPs. The specific models of the CPs and SPs are shown in Figures 2 and 3.

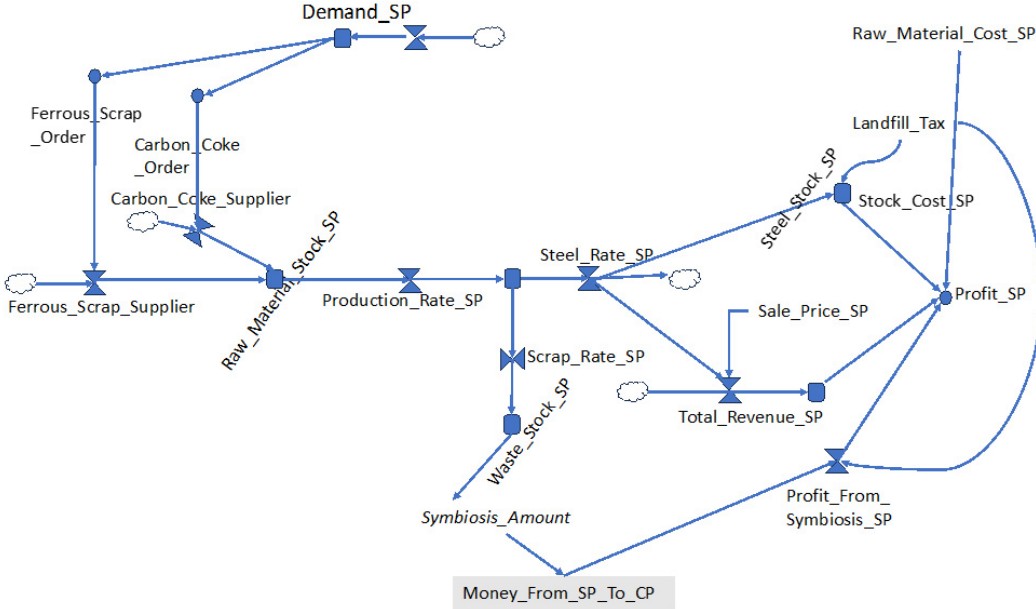

**Figure 2.** Model of SP using the SD approach.

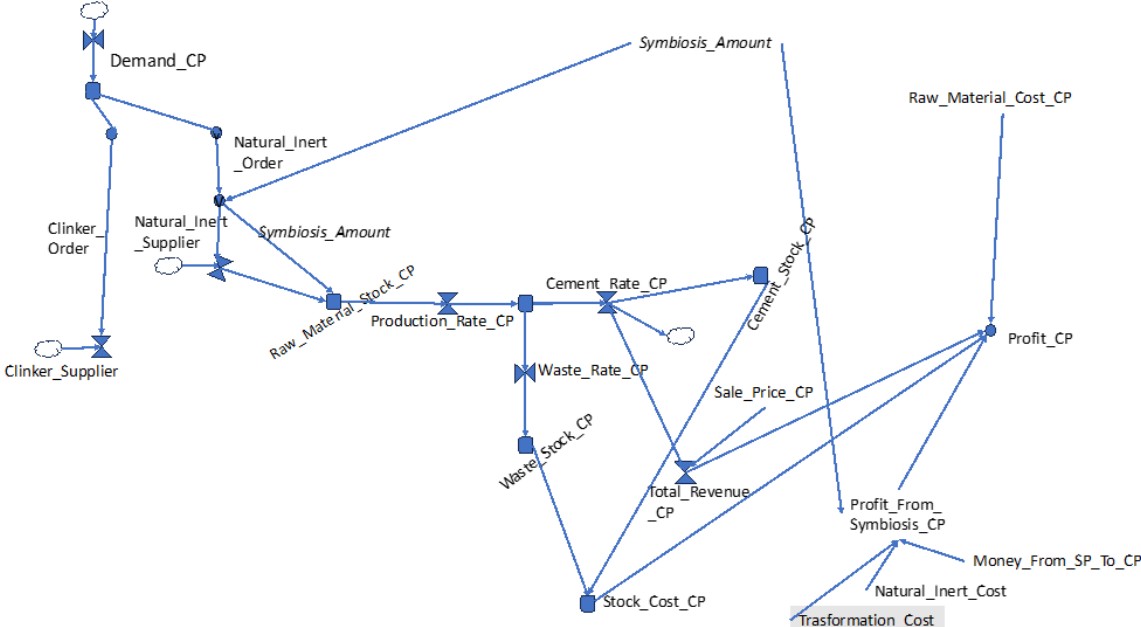

**Figure 3.** Model of CP using the SD approach.

The demand for SP and CP is stochastic: it is extracted from a normal distribution with a mean value equal to 20,000 ton/week.

Referring to Figure 2, as a function of the extracted value for steel demand, the ferrous scrap and carbon coke orders are forwarded to the respective suppliers. The ordered quantities are then dispatched and flow into the stock called Raw_Material_Stock_SP. When raw materials are available, steel production begins. The production process is managed by the flow Production_Rate_SP. The production process results in the finished product (steel) and scrap. The last one is directly related to the stock of waste, which will be the material of exchange in the industrial symbiosis process. For SP, the profit relative to IS is a function of both *Landfill_Tax* and *Money_From_SP_To_CP* (highlighted in grey in Figure 2). The value of this last variable depends on *Unit_Money_From_SP_To_CP*, whose value is made to vary iteratively according to what is described in Section 3.3.

Referring to Figure 3, as a function of the extracted value for cement demand, the clinker order and natural inert materials order are assessed. With regard to the *Natural_Inert_Supplier*, it is evaluated according to the possibility of having a feasible IS relationship with SP (Equations (5) and (6)). The ordered quantities are then dispatched and flow into the stock called Raw_Material_Stock_CP. When raw materials are available, cement production begins. The production process is managed by the flow Production_Rate_CP. The production process determines two flows: one of the final products and the other directed to the waste stock. The final price of cement depends on the type of raw materials used. For CP, the profit relative to the IS is a function of *Money_From_SP_To_CP*, *Natural_Inert_Cost*, and *Transformation_Cost*. The value of this last variable (highlighted in grey in Figure 3) depends on transport cost according to Equation (3). The cost of transport, in turn, depends on the transport system used, which is varied according to what is described in Section 3.3.

### 3.1. The Model KPIs

The performance of IS relationships will be evaluated through a sustainable paradigm [24–26] specifically as described in the following Section.

The economic dimension is weekly assessed by calculating the total profit after implementing an IS network, if an IS relationship is feasible. In this regard, the amount of additional profit a plant could obtain through the IS would be calculated by Equations (13) and (14).

$$Profit\_From\_Symbiosis\_SP = Landfill\_Tax - Money\_From\_SP\_To\_CP \tag{13}$$

$$Profit\_From\_Symbiosis\_CP = Natural\_Inert\_Cost - (Transformation\_Costs - Money\_From\_SP\_To\_CP)) \tag{14}$$

The environmental dimension is weekly assessed in terms of $CO_2$ emissions, which depend on the transport system and other factors such as distance. $CO_2$ emissions could be calculated through Equation (15), according to "Guidelines for Measuring and Managing $CO_2$ Emissions from Freight Transport Operations" published by ECTA and Cefic. Emissions are only proportional to the travelled kilometers and to the load on the trucks (*Symbiosis_Amount*) and not to the travel time, therefore congestion on the roads does not affect $CO_2$ emissions.

$$CO_2\_Emissions_m = \sum_{\forall v \in m} Travelled_{Tons\,v} \times Travelled_{Km\,v} \times \left( \frac{g\ CO_2}{t \times Km} \right)_v \tag{15}$$

where the summation is extended to all vehicles $v$ in the transport system $m$. The transport systems analyzed are described in detail in Section 3.2.

The social dimension is weekly valued through two alternatives: creation of job positions from the profit of symbiosis and investing the profit from symbiosis money in social activities for the benefit of workers and environmental activities. In the first case, the job created is a truck driver. A job is assumed to be created every 300 km traveled per week. The number of jobs created is calculated through Formula (16) and sub-formulas (16a) and (16b). In these formulas, the function "E" represents the integer part of the value, and it is assumed that the return journey has the same length as the outward journey. As it concerns the amount of money invested in training and other activities for the benefit of workers, it is decided that 20% of each company's symbiosis profit is used for this aim.

$$Number\_Of\_Created\_Jobs = E\left(Travelled\_Km_{truck}/300\right) \tag{16}$$

$$Travelled\_Km_{truck} = Number\_Of\_Trips_{truck} \times 2 \times Trip\_Lenght\_Km_{truck} \tag{16a}$$

$$Number\_Of\_Trips_{truck} = E\left( \frac{Symbiosis\_Amount}{Capacity_{truck}} \right) \tag{16b}$$

*3.2. Different Transportation Systems: The Simulation Scenarios*

Three scenarios have been considered, each scenario referring to a different transportation system.

The transportation system may involve a single vehicle (in this case, a truck) or two different vehicles (in this second case: a truck from SP to the train station, rail transport between two train stations, and another truck again from the train station to CP). In bi-modal transport, the trucks are of the same type.

The transportation system significantly impacts the freight transport costs, which, in turn, affect the *Transformation_Costs*. Therefore, the transportation system plays a crucial role in the feasibility of IS and has substantial effects on the economic and environmental dimensions of IS KPI.

A comprehensive transport cost evaluation model includes the following cost components:

- **The cost of fuel:** It varies depending on the type of fuel and the region. However, it generally accounts for 25–30% of the total cost of freight transport.
- **Drivers' wages:** typically, they account for about 30–35% of the total cost of freight transport.
- **Maintenance and repairs:** this includes routine maintenance but also unforeseen repairs and represent typically about 10–15%.
- **Equipment costs:** this includes the costs of purchasing and maintaining the truck and trailer, which typically account for about 10–15% of the total transport cost.
- **Insurance and administrative costs:** this includes aspects such as dispatching, billing, and tracking.

The cost of fuel was considered in the research. It is assessed on a weekly basis and is proportional to both the cost of a unit of fuel and weekly fuel consumption, which, in turn, is proportional to the kilometers travelled during the week. The assumed value for the cost of a unit of fuel is 1.6 EUR/liter [27]. As for the weekly travelled kilometers, the following assumptions have been made.

- The route between an origin–destination pair is always the minimum distance route. There could be alternative faster routes than the shortest distance one if there was congestion on the network, but this was not considered.
- The number of trips between an origin–destination pair made per week depends on the *Symbiosis_Amount* between the origin–destination pair: a trip is made only when there is enough *Symbiosis_Amount* to have full load truck.

Drivers' wages, maintenance and repairs, and equipment costs were considered in the research. Insurance and administrative costs have been considered negligible components since the material transported is waste.

In Tables 4 and 5, the value corresponding to: the "Cost" column and the "Full load Truck" row represents the *Unit_Transport_Cost*$_v$ for the specific vehicle *v*. The value corresponding to: the "Cost" column and in the "Empty" row contributes, together with the trip length, to the assessment of the *Empty_Trip_Cost*$_v$ for the specific vehicle *v*. Return trips are assumed to be performed with empty vehicles.

**Table 4.** Traditional trucks features.

| Load Status | CO$_2$ Emissions | Fuel Consumption | Cost |
|---|---|---|---|
| Full load truck | 900 g CO$_2$/km | 39.2 lt per 100 km | 0.6000 EUR/(ton·km) |
| Empty | 773 g CO$_2$/km | 29.3 lt per 100 km | 0.451EUR/km |

**Table 5.** Sustainable truck characteristics.

| Load Status | $CO_2$ Emissions | Fuel Consumption | Cost |
|---|---|---|---|
| Full load truck | 45 $CO_2$/km | 33.32 lt per 100 km | 0.51 EUR/(ton·km) |
| Empty | 38.65 $CO_2$/km | 24.905 lt per 100 km | 0.386 EUR/km |
| Compared with traditional truck | 95% less | 15% less | 15% less |
| Average $CO_2$ emissions for a full truck | 18 g per ton per km (website) | | |

The transportation system impacts also on the environmental dimension of IS KPI. Below, for each transportation system, the grams of $CO_2$ produced by the mode of transport per unit of mass transported per 1 km have been reported.

3.2.1. TT Scenario: Traditional Truck Scenario

This scenario refers to a single modal transport system that involves traditional trucks. The vehicle loading capacity is 40 tons. The features of traditional trucks are reported in Table 4. Fuel consumption and $CO_2$ emissions have been assessed according to the comparative analysis of energy consumption and $CO_2$ emissions of road transport and combined transport road/rail available online [28].

3.2.2. SS Scenario: Inter Mode Scenario

This scenario refers to a single modal transport system that involves IVECO STRALIS NP 460. The vehicle loading capacity is 40 tons. The maintenance interval is 90,000 km. The features of sustainable trucks are reported in Table 5. Reference for fuel consumption and $CO_2$ emissions is the Iveco New Stralis NP 460 website [29].

3.2.3. IM Scenario: Inter Mode Scenario

This scenario pertains to a bi-modal transportation system: sustainable trucks (the one considered in scenario SS) combined with trains capable of transporting 5000 tons per trip. The transportation process in the IM can be resumed in the following three steps:

- 1st step: transportation by sustainable trucks from SP to the railway station, with full truck trips.
- 2nd step: railway transportation; all the weekly waste is transported in one trip.
- 3rd step: waste transportation by trucks from the railway station to CP, using full truck trips.

In this scenario, both companies should be equipped with trucks to transport the waste between the plant and the train station. The train characteristics are reported in Table 6. Assuming a currency exchange rate of USD 1 = EUR 0.85, the train costs have been obtained from cost-per-ton-mile-by-mode-of-transportation available online [30].

**Table 6.** Train characteristics.

| Load Status | Cost |
|---|---|
| Full load train | 0.04 EUR/(ton·km) |
| Empty train | 0.386 EUR/km |
| Average $CO_2$ emissions for an empty train | 15 g per ton per km |

*3.3. The Iterative Process*

Each feasibility check is performed for each link ij, beween $CP_i$ and $SP_j$, under specific conditions that refer to *Money_From_SP_To_CP* and *Transport_Cost$_m$*. These conditions are varied iteratively, according to Figure 4. At each iteration, *Unit_Money_From_SP_To_CP* is decreased, starting from a maximum value down to a critical value for which the IS relationships result unfeasible. For a given value for *Unit_Money_From_SP_To_CP*, the

system dynamic approach performs the feasibility check of the IS relationship for each of the three proposed transport systems, to which a specific $Transport\_Cost_m$ corresponds. If the IS relationship results feasible, the link between the two nodes/agents is confirmed for the current week and the system dynamic model allows for the assessment of the $Symbiosis\_Amount$ (according to Equation (10)) and the related KPIs, as described in Section 3.1. If the IS relationship is feasible for at least one transport system, it is further reduced $Unit\_Money\_From\_SP\_To\_CP$ (k: = k + 1) and the feasibility of IS is reverified. If the IS relationship is not feasible, the link between the two nodes/agents will be removed.

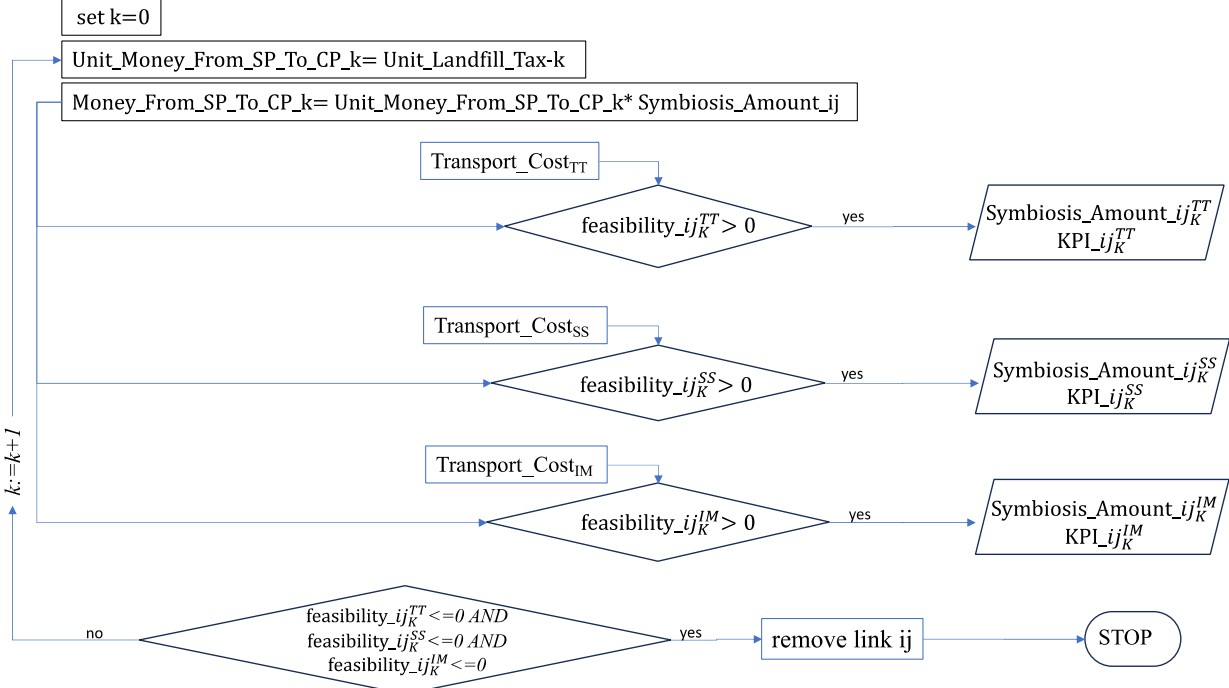

**Figure 4.** Flowchart of the iterative process.

## 4. Results

The proposed hybrid model was applied to a realistic case study to assess the feasibility of potential IS relationships between plants in the area on a weekly basis, considering the availability of three different transportation systems. The case study involves 14 CPs and 14 SPs in the Italian territory. Figure 5 shows the geographical locations of these plants. The longitude and latitude values of CPs and SPs are provided in Table A3 in Appendix B.

The simulation period spans 52 weeks, equivalent to one year. The landfill tax value ($Unit\_Landfill\_Tax$) is assumed to be 50 EUR/ton across the entire territory.

Over the 52-week period, the system's behavior was simulated, specifically focusing on the activation or deactivation of IS relations within the network on a weekly basis. For each week, each IS relation was tested with different values of $Unit\_Money\_From\_SP\_To\_CP$ and for the three transport modes, as outlined in Section 3.3. $Unit\_Money\_From\_SP\_To\_CP$ is decreased from 50 EUR/ton to 26 EUR/ton, at which point the IS relation is no longer feasible with any transport system. For a specific value of $Unit\_Money\_From\_SP\_To\_CP$ the IS relation could be feasible with one or more transport modes. In such cases, the simulator provides KPI values for each mode. Naturally, it is possible that an IS relationship is not feasible with any transport mode during a given week.

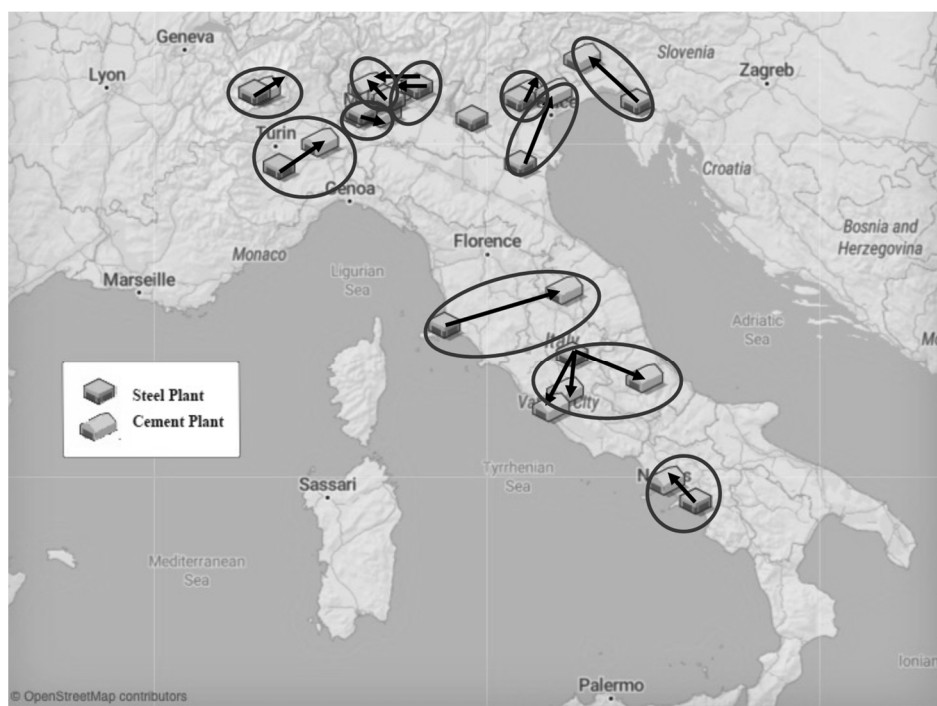

**Figure 5.** Geographical location of SPs and CPs in the case study. IS relationships being active (feasible) at week 15, *Unit_Money_From_SP_To_CP* = 46 EUR/ton.

For the sake of simplicity, the following results are presented for week 52 alone, representing the IS relations that emerged throughout the entire simulation period. Particularly, the results related to *Unit_Money_From_SP_To_CP* = 46 EUR/ton are shown in Table 7. For this value, both the SS and IM scenarios enable IS feasibility for certain links within the network. The reported results are aggregated as they pertain to all SP plants and all CP plants for which IS relationship results feasible. It should be noted that when *Unit_Money_From_SP_To_CP* is 46 EUR/ton, IS feasibility is denied for all the links in the case of TT scenario.

**Table 7.** Results of hybrid model, SS, and IM scenarios, week 52, *Unit_Money_From_SP_To_CP* = 46 EUR/ton.

| | KPIs | SS Scenario | IM Scenario |
|---|---|---|---|
| Steel Plant (SP) | Profits from Symbiosis [EUR] | 310,938 | 390,136 |
| | Total profits [EUR] | 213,099,808 | 212,583,673 |
| Cement plant (CP) | Profits from Symbiosis [EUR] | 4,526,726 | 4,526,726 |
| | Total profits [EUR] | 58,700,329 | 60,045,153 |
| | Transportation costs [EUR] | 2,013,577 | 1,169,694 |
| | $CO_2$ emissions [ton] | 174 | 89 |
| | Money invested for training and entertainment activities [EUR] | 905,345 | 897,864 |
| Material exchange | Amount of symbiosis [ton] | 77,735 | 97,534 |
| | Total inert materials used (artificial + natural) [ton] | 845,273 | 863,041 |

By further reducing the value of the *Unit_Money_From_SP_To_CP* from 46 EUR/ton, IS relationships are no longer feasible in the SS scenario, while IS relationships remain feasible in the IM scenario until the value of *Unit_Money_From_SP_To_CP* reaches 26 EUR/ton. Below this value, IS relationships are not feasible for any scenario. This highlights the advantages of the IM scenario. Naturally, as the amount of money transferred from SP to CP increases, the profit from symbiosis for SP decreases while it grows for CP. To ensure a

fair sharing of wealth between the different companies, it is recommended to apply the case where the amount of money given by SP to CP is 26 EUR/ton.

Table 7, which corresponds to a *Unit_Money_From_SP_To_CP* of 46 EUR/ton, reveals that the level of symbiosis in the SS scenario is 77,735 tons, while in the IM scenario, it is 97,534 tons. This discrepancy is primarily due to the significantly lower transportation costs in the IM scenario, with EUR 2,017,577 in SS compared to EUR 1,169,694 in IM. Consequently, relying solely on sustainable trucks as the mode of transportation results in relatively modest KPI values, as the amount of symbiosis in the SS scenario is 20% lower than the amount of symbiosis in the intermodal approach. Concurrently, the IM scenario boasts a higher volume of waste exchange, leading to superior KPIs compared to the other scenarios.

Figure 5 shows the active IS relations in week 15 with *Unit_Money_From_SP_To_CP* set at 46 EUR/ton. Figure 6 pertains to the simulation results that emerged at the conclusion of the simulation period, which is week 52, still maintaining *Unit_Money_From_SP_To_CP* at 46 EUR/ton. In Figure 6, IS relations that remain active at the end of the simulation period are indicated by the thick arrows. It is worth noting that the clusters in Figures 5 and 6 are different. Each cluster comprises one SP and one or more CPs. For all CPs within each cluster, the simulation established feasible IS relationships with the SP in the same cluster, using one or more transport systems. Plants that do not belong to any cluster were unable to establish feasible IS relationships during the simulation. When comparing Figures 5 and 6, transitioning from week 15 to week 52, five links were discontinued. In particular, links longer than 64 km were eliminated. This suggests that for shorter connections, i.e., when the distance between the SP plant and the CP plant is shorter, there is a greater likelihood that the IS relationship will remain feasible in the long run.

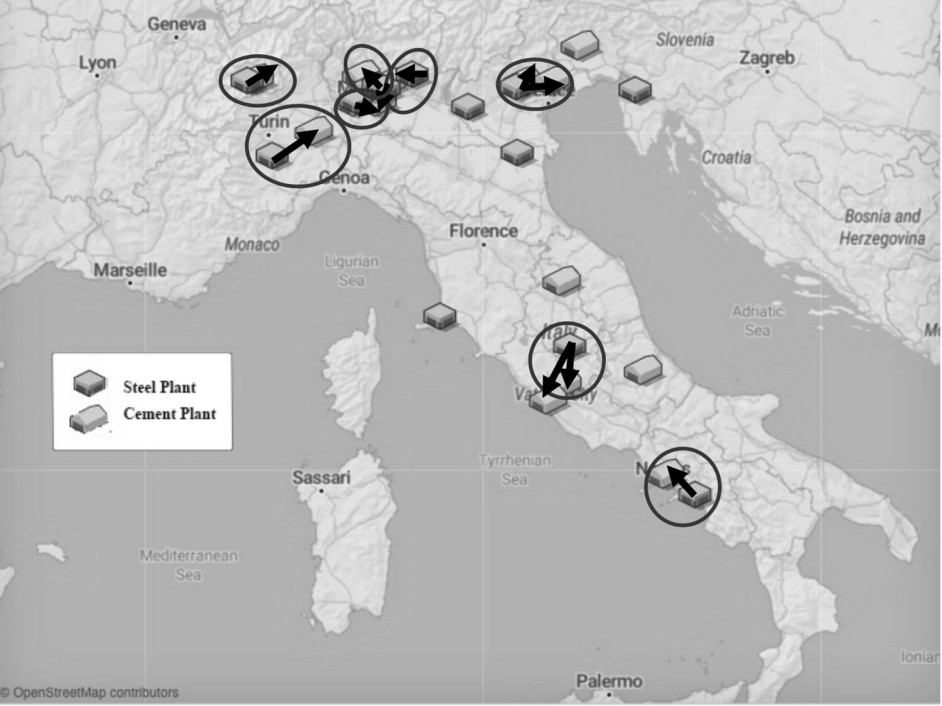

**Figure 6.** IS relationships remaining active (feasible) at the end of the 52-week period, *Unit_Money_From_SP_To_CP* = 46 EUR/ton.

Table 8 provides a comparable overview of KPIs in terms of profit and $CO_2$ emissions. The best performance in terms of profit is associated with the IM scenario, as the symbiosis profit for SP is slightly higher in the IM scenario compared to the SS scenario (EUR 390,136 against EUR 310,938), while the profits for CP remain the same in both scenarios. In terms of percentages, the symbiosis profit as a proportion of the total profit for SP is 0.15% in

the SS scenario, whereas it increases to 0.18% in the IM scenario. This can be attributed to the fact that *Unit_Money_From_SP_To_CP* is closely aligned with the landfill tax value (50 EUR/ton). In terms of $CO_2$ emissions, the SS scenario results in emissions of 174 tons, while the IM scenario produces emissions of 89 tons. Consequently, utilizing an intermodal transport system reduces $CO_2$ emissions by approximately 50% when compared to the SS scenario.

**Table 8.** Results, in percentage terms, of the hybrid model, SS, and IM scenarios, week 52, *Unit_Money_From_SP_To_CP* = 46 EUR/ton.

| | KPIs | SS Scenario | IM Scenario |
|---|---|---|---|
| | SP: Profits from symbiosis/total profits | 0.15% | 0.18% |
| | CP: Profits from symbiosis/total profits | 7.7% | 7.5% |
| Symbiosis_Amount/Total_Inert | | 9.20% | 11.30% |
| | $CO_2$ emissions_IM/$CO_2$ emissions_SS | | 50% |
| Transport_Cost_IM/Transport_Cost_SS | | | 58% |

### 4.1. Sensitivity Analysis

Section 4 presents the results related to the application of the proposed hybrid model in a predefined network with fixed and known distances between plants. In this network, *Unit_Money_from_SP_To_CP* was varied, following the flowchart in Figure 4, ranging from 46 EUR/ton down to 26 EUR/ton. *Unit_Landfill_Tax* was assumed to be 50 EUR/ton. We simulated the adaptive behavior of the system over 52 weeks. Each week, we assessed the IS feasibility of each link, and if necessary, we calculated the symbiosis amount and related KPIs.

This Section introduces a sensitivity analysis for the system dynamic component of the hybrid model. For a constant value of *Unit_Money_from_SP_To_CP*, set at 26 EUR/ton, we gradually increased the distance between a single pair of plants. *Unit_Landfill_Tax* remained at 50 EUR/ton. For each distance value and for each of the three transport system scenarios, we assessed the IS feasibility of the link. The process concluded when we identified a distance value at which the IS relationship was no longer feasible for any mode of transport. This approach allowed us to determine the critical distance in each scenario, which is the distance beyond which the IS relationship is no longer feasible for a given mode of transport. The simulation spanned one week, and the results are presented in Figure 7.

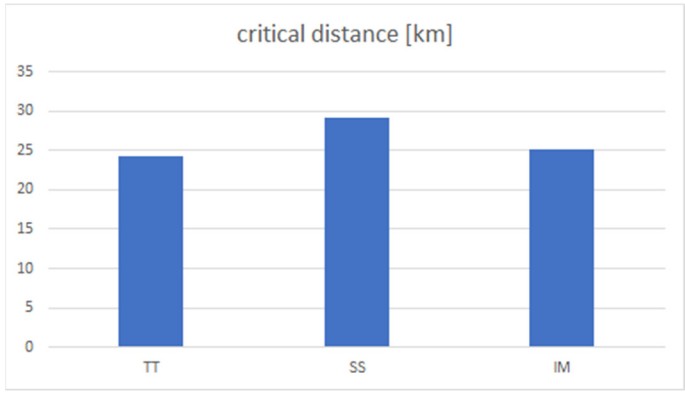

**Figure 7.** Critical distance for the three scenarios, *Unit_Money_from_SP_To_CP* = 26 EUR/ton.

The simulations reveal that the critical distance for the TT scenario is 24.6 km, for the SS scenario it is 29.3 km, and for the IM scenario, it is 25.1 km. Symbiotic relationships remain viable over longer distances when the mode of transport is SS, extending up to nearly 30 km,

with *Unit_Money_from_SP_To_CP* set at 26 EUR/ton and *Unit_Landfill_Tax* at 50 EUR/ton. It is worth noting that in the intermodal scenario, only the distance between the plants and the railway stations (the route traveled by the truck) has been increased, not the distance between the two stations; the distance traveled by the train remains constant at 24 km. The results obtained are, therefore, not surprising, as rail transport is not cost-effective for short distances.

## 5. Discussion and Conclusions

Establishing IS relationships among plants necessitates meeting a series of conditions to ensure the profitability of participating plants. One of the primary factors in initiating the exchange of materials among plants is the mode of transportation, which has often been overlooked in previous studies. To address this, the paper introduces a hybrid model that allows for the synthetic modeling of production processes through an SD approach. Simultaneously, it employs an AB approach to capture the adaptive behavior through which plant agents establish IS relationships over time. The feasibility of each IS relationship is further influenced by the transportation modes available for the exchange of materials, with three different scenarios for three modes of transportation introduced.

The proposed framework and simulation approach have been applied to a realistic case study, encompassing 14 CPs and 14 SPs in the Italian territory. The study has been analyzed within the context of the transportation dimension, simulating traditional, sustainable, and intermodal scenarios. The results of the study proved the following.

a. Distance plays a crucial role in establishing stable IS networks. This is evident from the results shown in both Sections 4 and 4.1. Referring to Section 4, in the long run, only IS relationships between neighboring plants endure. IS relationships on longer links have died out over time: links longer than 64 km have died out. This result aligns with what was found by Jensen et al. [31] that analyzed the proximity of symbiotic companies in the United Kingdom during their first five years of operation. The critical role of the proximity of participating companies in IS is also emphasized by Patricio et al. [32]: these authors proposed a method for identifying IS opportunities and, among the restrictions they applied, one related to linear distances between potential donors and receivers: they excluded facilities that were more than 48 km apart. However, for values close to 50 km, in the proposed case study, IS relationships are still feasible in scenarios involving sustainable transport systems. This difference between the values of the critical distance (64 km compared to 48 km) could depend on the different values of the parameters (including those related to transport systems) considered in the two researches, but it is certainly also related to the fact that in the proposed research the distances are real, considering the shortest path connecting each pair of plants, while in [32] the distances are linear and therefore shorter. Thus, a critical linear distance of 48 km could correspond to a real distance of 64 km. Furthermore, the sensitivity analysis presented in Section 4.1 shows that as the distance between plants increases, IS relationships exist in fewer and fewer scenarios. For distances of less than 24 km, the relationship is feasible in all scenarios; when the distance increases to 24.2 km, the relationship is no longer feasible in the TT scenario while it persists in the other two scenarios. As the distance increases further, the feasibility is lost even in the IM scenario. Finally, as the distance between the plants increases again, when this exceeds 29 km, the relationship is no longer feasible in any scenario. The results of the feasibility analysis align with the trends observed in the literature and the findings presented in Section 4. However, it is worth noting that the specific critical distance values observed in our study are not directly corroborated by the existing literature. These critical distances are contingent on specific parameter values, including a notably low *Unit_Money_from_SP_To_CP* value of 26 EUR/ton.

b. The utilization of sustainable of transportation modes enhances the possibility of feasible IS relations (as described in I) and the benefits of these (as described in II).

I.　　The results, presented in Section 4 for week 52, with *Unit_Money_From_SP_To_CP* set at 46 EUR/ton, demonstrate that the use of traditional transport modes (TT) does not lead to the feasibility of any IS relations within the analyzed network. In contrast, some relations are feasible when employing more sustainable transport modes (SS and IM). The sensitivity analysis in Section 4.1 further underscores that IS relations are only feasible on longer links in the SS scenario. This observation aligns with [33], which suggests that since waste is mostly of low economic value, transport and environmental costs may compromise symbiotic relationships, particularly over extended distances. Therefore, sustainable modes, which entail lower transport and environmental costs, enhance the feasibility of IS relationships.

II.　　Regarding the benefits, once again referencing the results for week 52 with *Unit_Money_From_SP_To_CP* set at 46 EUR/ton, Tables 7 and 8 illustrate that the IM intermodal scenario yields several advantages: a higher symbiosis profit for SPs, a higher total profit for CPs, lower transport costs, and a reduction of about 50% in $CO_2$ emissions compared to the SS scenario. These findings suggest that while feasible IS relations exist in both the SS and IM scenarios, the use of intermodality in the analyzed case leads to greater benefits in terms of both profit and transportation. Since waste transport plays a significant role in secondary emissions within industrial symbiosis [34], the adoption of low-emission transport modes enhances the environmental benefits of IS. Furthermore, the IM scenario results in higher profits due to the greater volume of material ex-changed, which leads to economies of scale and subsequently lowers unit processing and transport costs. This outcome aligns with research by Yu et al. [35], demonstrating a positive correlation between changes in unit transportation cost, unit processing cost, unit raw material cost, and unit waste disposal cost with economic objectives. As unit transportation cost and unit processing cost increase, economic IS benefits tend to decrease.

Hence, it is crucial to employ sustainable modes of transportation to mitigate emissions that significantly affect IS KPIs. Furthermore, an additional reduction in $CO_2$ emissions can be attained by minimizing the transportation distance between the plants [36].

Transportation is, therefore, a critical factor in establishing and sustaining an industrial symbiotic network.

As we have observed from the general transport cost assessment model, drivers' wages account for approximately 30–35% of the total cost of freight transport. It would be highly intriguing to explore the impact of autonomous vehicles on the feasibility of IS relationships. Similarly, as we have discerned from the general transport cost assessment model, the cost of fuel affects about 25–30% of the total cost of freight transport. The outcomes of the proposed analyses are closely tied to the geographical locations of the plants and the specific historical context to which the analysis pertains. These factors must be considered when extrapolating the results to different situations.

As future perspectives, the following remarks could be followed:

- Having a cluster in the industrial cities where the plants are near each other to reduce transportation costs. This cluster consolidates the circular economy.
- Having a company responsible only for logistics to transport the waste between the different plants, promoting intermodal transport with a fleet of sustainable trucks.
- Implementing a constraint on $CO_2$ emissions to set limits that transport systems should not exceed, considering potential policy interventions aimed at fostering the adoption of IS [37,38].

**Author Contributions:** Conceptualization and methodology: F.T.; software: M.A.A.; formal analysis and validation: F.B.; formal analysis and investigation and data curation: E.M.C.; writing—original draft preparation: F.B.; writing—review and editing: E.M.C.; supervision: F.T. All authors have read and agreed to the published version of the manuscript.

**Funding:** This research received no external funding.

**Data Availability Statement:** Data are contained within the article.

**Conflicts of Interest:** The authors declare no conflict of interest.

## Appendix A

The details about parameters of CPs and SPs are presented in Tables A1 and A2.

**Table A1.** Parameters for SPs.

| Parameters | Value | Unit |
|---|---|---|
| Necessary ferrous scrap to produce a unit of steel | 1.09 | Coefficient |
| Necessary carbon coke to produce a unit of steel | 0.01 | Coefficient |
| Production capacity | 23,000 | ton/week |
| Ferrous scrap price | 250 | EUR/ton |
| Carbon coke price | 150 | EUR/ton |
| Other costs (Labor, energy, maintenance, storage, etc.) | 500,000 | EUR/week |
| Waste per unit | 0.0147 | Coefficient |
| Sales price per unit | 500 | EUR/ton |
| Landfill Tax | 50 | EUR/ton |
| Money given from the SP to CP | 26 | EUR/ton |
| Preprocessing cost for waste | 20 | EUR/ton |

**Table A2.** Parameters for CPs.

| Parameters | Value | Unit |
|---|---|---|
| Clinker to produce a unit of cement | 0.316 | Coefficient |
| Necessary natural inert materials to produce a unit of cement | 0.74 | Coefficient |
| Production capacity | 23,000 | ton/week |
| Clinker price | 20 | EUR/ton |
| Natural inert materials price | 16 | EUR/ton |
| Other costs (Labor, energy, maintenance, storage, etc.) | 350,000 | EUR/week |
| Waste per unit | 0.047 | Coefficient |
| Sales price per unit | 85 | EUR/ton |
| Landfill Tax | 50 | EUR/ton |

## Appendix B

Regarding the application of the proposed methodology to the case study, Table A3 shows the longitude and latitude values of the SPs and CPs.

**Table A3.** The geographical location of SP and CP.

| Steel Plant | | Cement Plant | |
|---|---|---|---|
| Latitude | Longitude | Latitude | Longitude |
| 45.50084 | 9.126976 | 45.12864 | 8.45174 |
| 42.56301 | 12.69253 | 45.7402 | 7.468337 |
| 44.89161 | 11.81765 | 45.69671 | 9.675868 |
| 45.7337 | 7.322638 | 45.78421 | 9.240603 |
| 45.64652 | 9.600882 | 45.47453 | 9.516486 |
| 45.6255 | 13.78252 | 45.47107 | 9.152712 |
| 40.71023 | 14.77827 | 40.95503 | 14.30427 |
| 45.66231 | 11.80128 | 43.34449 | 12.58057 |
| 45.6795 | 9.505228 | 42.24416 | 13.93675 |
| 45.42958 | 10.9979 | 46.12609 | 12.88378 |
| 45.64508 | 9.367607 | 41.88176 | 12.35752 |
| 45.80549 | 10.06762 | 42.08058 | 12.59295 |
| 42.93003 | 10.53169 | 45.72107 | 12.40429 |
| 44.85535 | 7.737723 | 45.70968 | 11.93029 |

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
