# Peer review of "Impact of Transportation Costs on the Establishment of an Industrial Symbiosis Network"

_sustainability, doi:10.3390/su152215701_

Round 1
Reviewer 1 Report
Comments and Suggestions for Authors
Minor Comments
- I don't see it necessary to put circular economy between quotation marks. A similar comment is valid for other terms (CAS, for example).
- In line 54, either capitalize S in symbiosis or uncapitalize I in industrial. A similar comment is valid for other terms (Ferrous scrap and Carbon coke, for example). Review the manuscript and remove these inconsistencies.
- The paragraph from line 84 to line 106, citing some previous studies, is written in a monotonic way. I would rewrite it to "melt" with the rest of the section (synthesize the citations instead of just listing them).
- What do you mean by "An SP node can be the origin of several links while a CP node is the destination of only one link?", in lines 182-183. Is this an assumption of your model or it's an industrial requirement we observe in reality?
- You use the acronym KPI (Key performance indicator, I am guessing) without defining it.
Other Comments
- Equations (13) and (14) will be valid only if everything else is kept the same.
- Does the literature call equations (13) and (14) profits? I see them more as savings or changes in the profit (additional profit) if everything is kept the same.
- You considered benefits to the symbiosis using equation (16), is there any related cost apart from CO2 emission in equation (15) and the transportation costs?
- Did you perform any sensitivity analysis?
Comments on the Quality of English Language
Some minor edits.
Reviewer 2 Report
Comments and Suggestions for Authors
Very glad to review this paper (sustainability-2656738). Thanks for your waiting. In this paper, the authors presented a hybrid approach based on Agent-Based Modeling and System Dynamics to evaluate industrial symbiosis from a transportation point of view. Overall, due to the limited prior research, the research topic of this paper appears to be relatively novel. However, in terms of the content of the paper, it seems that your main contributions and efforts may not be particularly adequate. Additionally, there are issues with the paper's structure and details. I recommend carefully revising and adding more content to enhance the completeness of this paper.
Main problems:
i. The literature review is proposed to be presented as a separate section.
ii. When introducing the contribution of this paper at the end of the Section 1, what is the purpose of introducing the research conclusions of Demartini, Tonelli, and Govindan et al.? Isn't the similar content already explained before?
iii. The flowchart in Figure 4 is drawn in a non-standard manner. so it is recommended to modify it. What’s more, the interpretation of Figure 4 in Section 2.4 does not correspond exactly to the content of Figure 4.
iv. There are too many contents in Section 2, so it is suggested to establish a new section to divide them.
v. This paper only proves the feasibility of the established model through three scenarios, then how to prove its quality?
vi. How does this paper use the established model to solve the case? The process is not reflected.
vii. The innovation and contribution of this paper are not obvious, and the workload seems not to be enough from the current content.
Minor problems:
viii. Verify if Equation (8) and Equation (16.1) are correct.
ix. Figure 2 and Figure 3 appear quite cluttered, with unclear color distinctions and insufficient labeling. Furthermore, additional textual explanations are necessary, as they will help readers better understand this paper.
x. The title of Table 6 omits a period at the end. The same problem occurs on line 441 and 446.
xi. When Table B.1 is first mentioned in the first paragraph of the results section, it is suggested to be modified to Table B.1 in Appendix B.
xii. There is extra white space on page 12, line 347.
Comments on the Quality of English Language
The language level is clear.
Reviewer 3 Report
Comments and Suggestions for Authors
Manuscript number: sustainability-2656738
The paper endeavors to unveil a more profound understanding of Industrial Symbiosis (IS) modeling. The authors employ Agent-Based Modeling and System Dynamics to simulate the intricate web of interactions between companies and their consequent impact on transportation. The paper holds merit for both the academic and industrial sectors. However, I have several recommendations to enhance the paper's quality, as detailed below:
The introduction section is overly lengthy and could potentially bewilder readers due to the multitude of topics covered, including Circular Economy (CE), Complex Adaptive Systems (CAS), Industrial Symbiosis (IS), Waste Management, and Transportation. Furthermore, the objectives are somewhat unclear. I would suggest breaking the introduction into distinct subsections, each dedicated to highlighting a specific issue. Additionally, the authors should incorporate more relevant literature for each of these issues.
The paper fails to clearly establish the research gap. Readers should be provided with insights into the novelty of this work. While the introduction does contain a brief mention of literature (limited to just two paragraphs) primarily focused on the methods, it lacks references to literature related to Circular Economy (CE), Complex Adaptive Systems (CAS), and Industrial Symbiosis (IS).
It would be beneficial if the authors could furnish reasons for their selection of Steel Plants (SPs) and Cement Plants (CPs), particularly drawing from pertinent prior literature. This would lend credibility to their choice of these specific entities.
The development of the model lacks crucial information. For instance, there is no reference provided for parameters such as truck fuel consumption, CO2 emissions, or cost. Including these references would enhance the model's robustness and transparency.
In the discussion section, the concept of "distance" should be connected to the model results, and recommendations stemming from these results should be elaborated upon. This will provide a more comprehensive understanding of how the model's outcomes can inform practical applications.
The discussion should be firmly rooted in the model results and previous research findings. Readers should be guided through the authors' interpretations of the model outcomes in the context of existing literature, facilitating a deeper comprehension of the implications and contributions of the study.
Round 2
Reviewer 2 Report
Comments and Suggestions for Authors
You have modified the relevant content according to the modification suggestions, but I think there are still the following problems for the revised paper (sustainability-2656738-R2):
1. Figure 4 should not have been deleted, because the flowchart can help us better understand the content of the paper, so it is suggested to redraw it.
2. The added Section 4 has been emphasizing the difficulties of model verification, which is not of great value to the verification of model quality. Therefore, this newly added section does not play its due role and is actually meaningless.
3. Judging from the contributions you have described, the work done in the paper is still not enough. It is suggested to add sensitivity analysis to make the paper more substantial.
4. Figure 2 and Figure 3 still have the same problem of unclear color distinction and unclear content, which has not been corrected.
5. There is extra white space above line 342.
Comments on the Quality of English LanguageThe language level is clear.
Reviewer 3 Report
Comments and Suggestions for Authors
Manuscript number: sustainability-2656738 R1
Thank you for your revision. However, this comment is not responded.
"The discussion should be firmly rooted in the model results and previous research findings. Readers should be guided through the authors' interpretations of the model outcomes in the context of existing literature, facilitating a deeper comprehension of the implications and contributions of the study."
Round 3
Reviewer 2 Report
Comments and Suggestions for Authors
Thanks for the authors' revision. The paper has been improved much.
It is good.
Comments on the Quality of English Language
The only concern is to check the language issues. There are some grammar problems to deal with.
Author Response
the article was revised by an English language expert. thanks